## Research Article

ocean; marine; ecosystem; biogeochemistry; phenology

**Corresponding author:**
Scott C. Doney;
Email: scd5c@virginia.edu

# Quantifying seasonal to multi-decadal signals in coastal water quality using high- and low-frequency time series data

Emma I. Brahmey, Karen J. McGlathery and Scott C. Doney 

Department of Environmental Sciences, University of Virginia, Charlottesville, VA, USA

## Abstract

To inform water quality monitoring techniques and modeling at coastal research sites, this study investigated seasonality and trends in coastal lagoons on the eastern shore of Virginia, USA. Seasonality was quantified with harmonic analysis of low-frequency time-series, approximately 30 years of quarterly sampled data at thirteen mainland, lagoon, and ocean inlet sites, along with 4–6 years of high-frequency, 15-min resolution sonde data at two mainland sites. Temperature, dissolved oxygen, and apparent oxygen utilization (AOU) seasonality were dominated by annual harmonics, while salinity and chlorophyll-*a* exhibited mixed annual and semi-annual harmonics. Mainland sites had larger seasonal amplitudes and higher peak summer values for temperature, chlorophyll-*a* and AOU, likely from longer water residence times, shallower waters, and proximity to marshes and uplands. Based on the statistical subsampling of high-frequency data, one to several decades of low-frequency data (at quarterly sampling) were needed to quantify the climatological seasonal cycle within specified confidence intervals. Statistically significant decadal warming and increasing chlorophyll-*a* concentrations were found at a sub-set of mainland sites, with no distinct geographic patterns for other water quality trends. The analysis highlighted challenges in detecting long-term trends in coastal water quality at sites sampled at low frequency with large seasonal and interannual variability.

## Impact statement

Accurate monitoring of interconnected water quality variables such as temperature, salinity, dissolved oxygen, chlorophyll-*a*, and apparent oxygen utilization (AOU) is vital for tracking the status, health, and dynamics of coastal marine ecosystems. For example, warming water temperatures can enhance the frequency and severity of algal blooms (elevated levels of chlorophyll), leading to the formation of hypoxic and anoxic zones (low to no dissolved oxygen), and influencing the community metabolism (AOU, which depends on a balance of photosynthesis, respiration and ventilation). However, there are logistical and scientific challenges in maintaining consistent and adequate water quality monitoring. High-frequency in-situ measurements using automated sondes have a high temporal frequency (i.e., every 15 min) and are less labor intensive. However, due to power and maintenance demands, these are mostly confined to shore-based sites or more geographically limited and expensive coastal scientific moorings and cabled arrays. Automated instruments are also subject to sensor malfunction or biofouling, leading to a consequent loss of consistent time series without frequent upkeep. Longer term sites measured at a low frequency have higher geographic spread and are sampled using a manual sonde, water sampling and lab extraction methods typically at a lower temporal frequency (i.e., weekly to quarterly), but are limited to safe conditions for shore-based or boat sampling. In this study, we investigated if there are differences between seasonal harmonic elements of high- and low-frequency data over site types and how sampling frequency affected estimates of the magnitude and timing of the seasonal cycle using sub-sampling, that is, sites measured at a high-frequency subsampled at the rate of sites measured at a low-frequency. Additionally, using boot-strapping techniques, we explored how many years of simulated quarterly sampling would be needed to quantify the climatological seasonal cycle within specified confidence intervals. We hope that this research can guide water quality monitoring techniques and modeling at other coastal research sites in order to adequately observe ecosystems in a changing climate.


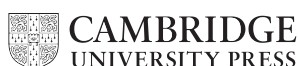

## Introduction

Coastal lagoons are typically shallow with submerged, often vegetated flats and intermittent openings to the sea (Scanes et al., 2007). The marshes, oyster reefs, and seagrass meadows in temperate coastal lagoons provide vital ecosystem services, including biodiversity, shoreline

protection, and carbon storage (Orth et al., 2020; Smith et al., 2022). These systems are vulnerable to environmental change, including habitat loss from coastal development, degradation of water quality from development and pollution, changes in biodiversity due to overexploitation, and invasive species, and climate change impacts, especially marine heat waves and sea-level rise (Newton et al., 2018). Water quality variables, such as temperature, salinity, dissolved oxygen (DO), apparent oxygen utilization (AOU), and chlorophyll-*a* (Chl), integrate responses to human perturbation and climate change and are monitored to document long-term change.

The Eastern Shore of coastal Virginia, USA, has an extensive coastal lagoon and barrier island system marked by a large expanse of relatively undeveloped rural coastline and lacks significant inputs of fluvial sources of freshwater and sediment (Safak et al., 2015). The lagoons studied here are on the eastern side of the Delmarva Peninsula, fronting the coastal ocean in the Mid-Atlantic Bight, with no direct hydrologic connection to the Chesapeake Bay estuary on the western side of the Peninsula. Historically, this area, the Virginia Coast Reserve (VCR), hosted large scallop fisheries due to the abundance of *Zostera marina* seagrass beds until a massive seagrass die-off event in the 1930s (Hondula and Pace, 2014; Oreska et al., 2017). A large-scale restoration effort by reseeding that began in 2001 has returned over 36 km$^2$ of seagrass as of 2021 (Orth et al., 2020; Oreska et al., 2021).

In the VCR coastal lagoons, water quality varies as a function of season, tidal currents and flushing, winds and storm conditions (Hondula and Pace, 2014). Climate is the most dominant driver of ecological change, especially sea-level rise, storms, increased temperature and marine heat waves (The Nature Conservancy, 2011; McGlathery et al., 2013). The VCR coastal lagoons have relatively low nutrient loading and water column chlorophyll concentrations (McGlathery et al., 2007; Carr et al., 2012), with little variation in salinity from adjacent coastal ocean waters due to limited freshwater discharge (Oreska et al., 2021). This good water quality largely reflects the low fluvial inputs, coastal development, and human influence on the VCR coastal lagoons. This differs from the more impaired estuary systems typical along the Mid-Atlantic seaboard, including the Chesapeake Bay, where substantial point source and non-point nutrient source pollution causes extensive coastal eutrophication and low-oxygen conditions (Sabo et al., 2022). The VCR coastal lagoons thus can serve, more generally, as an end-member for the pre-industrial low-disturbance or future recovery state of coastal water quality at other temperate monitoring and research sites.

The Virginia Coast Reserve Long-Term Ecological Research (VCR-LTER) project maintains thirteen long-term (multi-decade) water quality monitoring sites measured at a low frequency within the coastal lagoon system. The sampling network spans across environment types, water depths and residence times with respect to tidal flushing (Safak et al., 2018). Most of the VCR-LTER sites are relatively shallow and oligotrophic, with a mean semi-diurnal tidal range of 1.2 m (Oreska et al., 2021). The Virginia Institute of Marine Science (VIMS) at the Eastern Shore Laboratory (ESL) (referred to as ESL hereafter) recently established two water quality monitoring sites with high-frequency, automated sonde measurements; both sites are shore-based on the mainland side of the lagoon system.

Data analysis of seasonal patterns in multi-year water quality data with persistent gaps can be accomplished using harmonic analysis, a method that represents fluctuations in a time series from the sum of sine and cosine functions that have different frequencies (Wilks, 2011). Higher harmonics indicate higher frequencies, with the first harmonic representing in this study one full annual cycle

and the second harmonic representing the semi-annual cycle (Wilks, 2011). The relative importance of each of these harmonics can be quantified using their respective fractions of total variance captured in the harmonic analysis, where for most variables, the majority of variance is captured by the first one or two harmonics. This study focused on the first and second annual harmonics because they contained a majority of the variances. A composite harmonic, the sum of the first and second harmonics, allows an analysis of the relative importance of each harmonic in the resulting fitted curve and harmonic model elements. Deviations from fitted harmonic curves can indicate anomalies from the climatological seasonal cycle, deseasonalized data, which allow for documentation of long-term trends and sub-seasonal variability. Seasonal harmonic elements such as amplitude, phase shift and minimum/maximum values can reveal important information about seasonal cycles. Below, we use the VCR coastal lagoon data to illustrate the utility of harmonic seasonal analysis and data sub-sampling techniques to coastal water quality monitoring sites.

## Materials and methods

### Study area and data description

The study was conducted with VCR-LTER data from the Eastern Shore of Virginia, USA, along the mid-latitude western continental boundary of the North Atlantic (Figure 1). High-frequency ESL data (temperature, salinity, dissolved oxygen and chlorophyll-*a*) (Ross and Snyder, 2020; Ross and Snyder, 2023) were measured every 15 min using YSI EXO2 Multiparameter Sondes attached to dual land-based pumps at two creek sites Wachapreague (W) and Willis Wharf (WW); the dual pump intakes were switched and cleaned weekly, the flow cell wall lightly cleaned monthly, and the datasondes calibrated every 90 days using a calibration standard and KorEXO software (Ross and Snyder, 2023; Figure 1). The data range from March 25, 2016 to December 31, 2022 at Wachapreague and October 12, 2018 to December 24, 2022 at Willis Wharf, with large gaps in both data (Ross and Snyder, 2020; Ross and Snyder, 2023; Table 1). As part of standard data quality assessment-quality control (QA/QC) methods, suspicious data or outliers were removed, excluding points that were outside of ±1 standard deviation (derived from yearly statistics of raw data) from the preceding data point (Ross and Snyder, 2023).

Low-frequency VCR-LTER data (temperature, salinity and dissolved oxygen), spanning 20–30 years (McGlathery and Christian, 2022), were collected manually using a YSI Datasonde lowered from a small boat or shore at each of the 13 sites; the datasondes were calibrated quarterly to annually following the manufacturer recommended procedure using calibration standards. Discrete water samples (200 mL) collected for chlorophyll-*a* were filtered through Whatman GF/F filters (0.7 um pore size), extracted in the dark for 24 h (90% acetone), and concentrations measured using a bench-top Shimadzu 1280 spectrophotometer (McGlathery and Christian, 2022; Table 1). Minimal QA/QC methods were applied to the database version of VCR LTER data to remove obvious outliers and bad data points. Water residence times with respect to tidal flushing were estimated using a three-dimensional, finite-volume coastal ocean model (FVCOM) that was then validated with field observations (Safak et al., 2018). Before conducting the harmonic and trend analysis, an additional data screening was conducted on any remaining outliers in both the ESL and VCR data sets using Chauvenet's Criterion (Glover et al., 2011).

The base-10 logarithm ($\log_{10}$) of chlorophyll-*a* was taken in order to make the chlorophyll data more closely normally

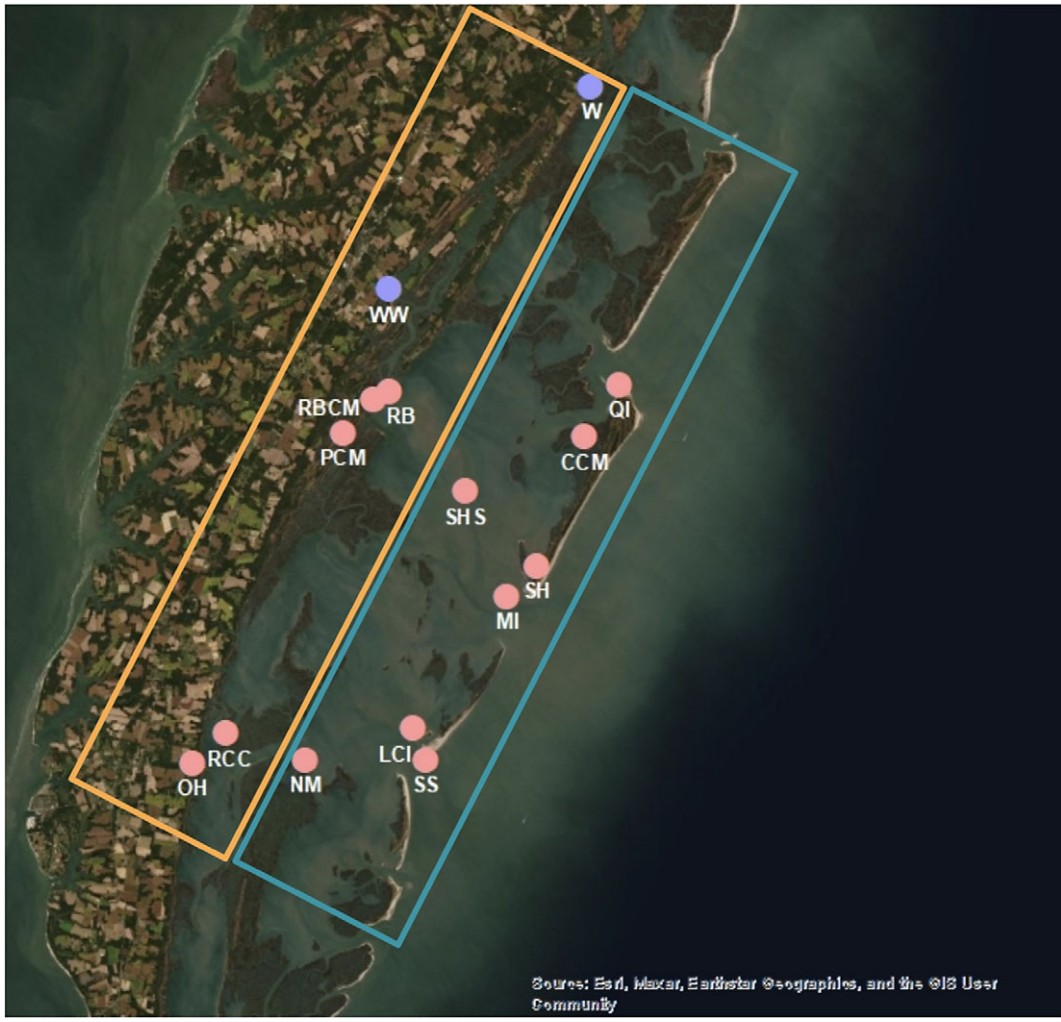

**Figure 1.** Spatial map of the eastern shore of Virginia, United States created in ArcGIS using Imagery (WGS84) base map showing the locations of the ESL sites as purple dots, and the VCR-LTER sites as pink dots (see Table 1). The sites in the orange box are considered mainland, and the sites in the teal box are considered ocean inlet and mid-lagoon sites (Table 1). The Virginia Coast Reserve shallow lagoon-barrier island system is bounded to the west by the Eastern Shore peninsula and to the east by barrier islands. The lagoon system is flushed by tidal flows from the coastal Atlantic Ocean (right side of image) via ocean inlets between the barrier islands.

distributed following typical practice for marine bio-optical data (Campbell, 1995). AOU was derived from temperature, salinity, and dissolved oxygen measurements and was calculated using Matlab code by Peltzer (2013) based on equations from Garcia and Gordon (1992). Because we lacked sufficient resolution of diurnal variability of photosynthesis and respiration for the sites of low-frequency data collection, no attempt was made to correct for possible aliasing of the diurnal cycle into VCR AOU estimates; daily averaging removed the diurnal cycle from the high-frequency ESL data.

### Data analysis

The water quality data were analyzed using a combination of harmonic analysis and linear regression with minimal a priori assumptions. Namely, the approach requires that a substantial fraction of the seasonal variability is captured by the annual and semi-annual harmonics and that residuals after removing the harmonics exhibit gradual trends that can be captured by a linear function (versus a step function, quadratic, etc.); both

assumptions hold reasonably well as shown in the Results and Discussion section.

Data for each site were compiled into climatological day of year graphs, and a harmonic model fit estimate, $y^m(t)$, was constructed using 1st and 2nd seasonal harmonics (sine curves) fit (Wilks, 2011):

$$y^m(t) = \bar{y} + \sum_k^2 \left( a_k \cos\left[\frac{2\pi kt}{n}\right] + b_k \sin\left[\frac{2\pi kt}{n}\right] \right), \qquad (1)$$

where $k$ was the respective harmonic, $\bar{y}$ was the mean of the $y$ values, $a_k$ and $b_k$ were the cosine and sine coefficients of the $k$th harmonic respectively, $t$ was day of year, $n$ was annual time period (365 days). Harmonic amplitude $(A_k)$, phase shift $(\phi_k)$, and date of maximum values $(t_k)$ were calculated using methods by Wilks (2011). Variances captured by the harmonic fit $(\sigma_k^2)$ and percent of the total fitted variance for each harmonic $(pV_k)$ were calculated using methods by Burroughs (2003). Confidence intervals and reduced chi-squared values $(\chi_v^2)$ were calculated using methods by Glover et al. (2011). Only statistically significant differences in harmonic parameters are highlighted in the Results and Discussion section.

**Table 1.** Names, site type, coordinates, and dates measured for each coastal water quality site in the Virginia Coast Reserve, where bolded sites are high-frequency ESL locations, and non-bolded sites low-frequency VCR-LTER locations

| Site | Site Type | Latitude (degree) | Longitude (degree) | Dates measured |
|---|---|---|---|---|
| **Wachapreague (W)** | Mainland | 37.608 | −75.686 | March 25, 2016 to December 31, 2022 |
| **Willis Wharf (WW)** | Mainland | 37.512 | −75.806 | October 12, 2018 to December 24, 2022 |
| Ramshorn Channel Creek (RCC) | Mainland | 37.303 | −75.905 | August 26, 2004 to October 11, 2022 |
| Redbank Creek Mouth (RBCM) | Mainland | 37.460 | −75.816 | August 26, 2004 to October 10, 2022 |
| Cattleshed Creek Mouth (CCM) | Ocean Inlet | 37.443 | −75.689 | July 28, 1992 to October 10, 2022 |
| Little Cobb Island (LCI) | Ocean Inlet | 37.305 | −75.792 | August 26, 2004 to October 11, 2022 |
| Machipongo Inlet (MI) | Ocean Inlet | 37.368 | −75.736 | July 31, 1997 to October 10, 2022 |
| New Marsh (NM) | Mid–Lagoon | 37.291 | −75.857 | August 26, 2004 to October 10, 2022 |
| Oyster Harbor (OH) | Mainland | 37.289 | −75.924 | July 28, 1992 to October 11, 2022 |
| Phillips Creek Mouth (PCM) | Mainland | 37.445 | −75.834 | July 28, 1992 to October 10, 2022 |
| Quinby Inlet (QI) | Ocean Inlet | 37.467 | −75.668 | August 25, 2004 to October 10, 2022 |
| Red Banks (RB) | Mainland | 37.464 | −75.807 | July 28, 1992 to October 10, 2022 |
| South Hog (SH) | Ocean Inlet | 37.382 | −75.718 | July 31, 1997 to October 10, 2022 |
| Shoal Site (SHS) | Mid–Lagoon | 37.417 | −75.761 | August 26, 2004 October 10, 2022 |
| Sand Shoal Inlet (SS) | Ocean Inlet | 37.290 | −75.785 | August 26, 2004 to October 11, 2022 |

The date of the seasonal minimum $(t_{min})$ and maximum $(t_{max})$ of the composite harmonics were computed using the zero-points of the first derivative with respect to time, $t$, of Equation 1. The amplitude of the composite harmonics was found from

$$A_{max} = \frac{y^m(t_{max}) - y^m(t_{min})}{2}, \qquad (2)$$

Bootstrapping was used to generate confidence intervals for the model parameters and resulting harmonic curves.

Generalized least squares (GLS) was applied to the deseasonalized data anomalies calculated using Equation 3 to see if there were any long-term trends for each of the sites and variables measured at low frequency.

$$y^{anom} = y_i - y^m(t_i), \qquad (3)$$

The long-term, low-frequency data sets were each broken at their midpoint into two equal-length halves of 7–15 years, depending on the dataset length, and separate harmonic curves were calculated for each half using Equation 1. The differences between the harmonic elements for the two time periods were calculated. The statistical significance of geographic and temporal differences in mean values was assessed, and $p$-values were reported using Student's $t$-test (Glover et al., 2011).

### *Simulated low-frequency sampling*

The ESL high-frequency time series were sub-sampled to create data sets with similar resolution as the VCR-LTER data to explore trade-offs among sampling frequency, duration, and climatological seasonal cycle resolution. The first set of experiments to evaluate the skill of the VCR-LTER data to resolve climatological cycles was conducted by randomly subsampling the daily averaged high-frequency data at the same sampling frequencies as the composite

climatological seasonal cycle for VCR-LTER mainland sites for each variable. The low-frequency sampling was conducted at a consistent point in the tidal cycle with the outgoing tide, and the daily averaging was applied to the high-frequency to minimize sub-diurnal tidal variability effects. Harmonic curves with confidence intervals for each parameter were computed using Equation 1 for 200 sub-sampled low-frequency series (trials) Equation 2. Root mean square error (RMSE) was computed between the low-frequency and high-frequency harmonic fits for each of the 200 trials with Equation 4.

$$RMSE = \sqrt{\frac{\sum_{i=1}^{365}\left(y_{full}(t_i) - y_{sub}(t_i)\right)^2}{n_{sub}}}, \qquad (4)$$

where $y_{full}(t_i)$ was the ith year day of the full, high-frequency harmonic fit (predicted value), $y_{sub}(t_i)$ was the subsampled, low-frequency harmonic fit value of that year day (trial value) and $n_{sub}$ was the variable-dependent number of subsampled values in the trial used as an estimate of degrees of freedom. RMSE was normalized (nRMSE) by dividing by the standard deviation of the deseasonalized anomalies from the full high-frequency harmonic data set calculated in Equation 3.

The Nash-Sutcliffe efficiency coefficient (NSE) is a metric for comparing goodness of fit across models with a maximum NSE = 1 (no error) and NSE = 0, indicating that the error variance is comparable to the observed variance (Nash and Sutcliffe, 1970).

$$NSE = 1 - \left(\frac{RMSE}{SD}\right)^2, \qquad (5)$$

where SD is the standard deviation of the randomly subsampled values for each trial.

NSE values were calculated for each variable to estimate the fit of the 200 randomly subsampled low-frequency trials versus the full

high-frequency data, and the average was taken. Ritter and Muñez-Carpena (2012) developed criteria to estimate the goodness of fit using this value, where NSEs above 0.90 are very good, 0.80–0.90 are good, 0.65–0.80 are acceptable, and below 0.65 are unsatisfactory.

A second set of experiments was performed to test the effect of time-series duration on resolving climatological cycles by varying the low-frequency sub-sampling rate for the Wachapreague and Willis Wharf site data sorted into seasons. Simulated time series were constructed by randomly subsampling the high-frequency data $m_{\text{sub−sample}} = 2 : m_y$ times per season (where $m_y$ is the maximum number of years for VCR time series). The $m_{\text{sub−sample}}$ value can be interpreted as the number of years (duration) of quarterly sampling or equivalently the duration multiplied by the quarterly sampling density. For each variable and $m_{\text{sub−sample}}$ value (seasonal sampling density), 200 simulated time-series (trials) were generated, harmonic analyses were performed for both the composite harmonic and dominant harmonic, and average values were computed for each harmonic fit parameter for the climatological seasonal cycle. RMSE and nRMSE for low-frequency versus high-frequency were calculated for each $m_{\text{sub−sample}}$ value using Equation 4 to assess error with $n_{\text{sub}} = 4m_{\text{sub−sample}}$.

Minimum sampling duration was determined by finding the lowest $m_{\text{sub−sample}}$ value where the sub-sampled composite harmonic elements for two sequential $m_{\text{sub−sample}}$ values fell within the high-frequency confidence intervals. In the special case where these intervals were not met, the closest value was selected, and that variable was noted. For the respective dominant harmonic, the standard deviations were calculated for the date of maximum and seasonal amplitudes from the 200 trials. These values were normalized (nSD) by calculating the effective asymptote with respective to high $m_{\text{sub−sample}}$ values, the average of the standard deviations of the 100th–200th trials.

## Results and Discussion

Coastal water quality metrics provide valuable information for assessing seasonal dynamics and trends in response to environmental drivers such as pollution and climate change (Tassone et al., 2021; Rosa et al., 2022; Buelo et al., 2024). In the lagoons of the Virginia Coast Reserve, strong seasonal patterns were seen in all water quality variables at both high- and low-frequency sampled sites.

### Temperature

For temperature, all sites were dominated by the first harmonic, indicating an annual climatological cycle primarily driven by meteorological factors and physical hydrological characteristics (Figure 2; Benyahya et al., 2007; Wiberg, 2023). Mainland sites' statistically significant earlier dates for summer peak temperatures (average of 213.6 ± 1.6 year day, $p = 0.0005$), and statistically significant higher maximum peak values (average of 29.21 ± 0.61°C, $p = 0.0005$) and seasonal amplitudes (average of 11.77 ± 0.08 °C, $p = 0.007$) were linked to their shallower depths that allowed for more rapid and intense temperature changes, while ocean inlet and mid-lagoon sites (average of 218.3 ± 2.4 year day, 27.51 ± 0.66 °C, and 11.26 ± 0.30°C) were dampened by routine exposure to cooler ocean water and shorter residence times (Safak et al., 2018; Supplemental Table S1).

### Salinity

For salinity, there was a mix of annual and semi-annual harmonics at the sites, differing by site geography, with yearly variations depending on the effects of mixing, evaporation, run-off, and precipitation (Sachithananthan, 1969). All of the mainland sites were dominated by the first harmonic, and most ocean inlet and mid-lagoon sites were dominated by the second harmonic (Figure 2). In general, ocean inlet and mid-lagoon sites showed less intra-site and temporal variability due to mitigation of the stronger impact of tidal flushing of the nearby ocean, which had an overall more stable seasonal cycle (Figure 2; NASA, n.d.). Peak salinity occurred at most sites mid-late year (mainland sites averaged 227.1 ± 27.0 year day, and ocean inlet and mid-lagoon sites averaged 219.0 ± 56.4 year day) (Supplemental Table S2). The mainland sites had more variable maximum values (average of 31.16 ± 0.76 psu) and statistically significant higher seasonal amplitudes (average of 1.61 ± 0.84 psu, $p = 0.036$), likely related to their differences in freshwater input and/or longer residence times, while ocean inlet and mid-lagoon sites had consistently higher maximum values (average of 31.51 ± 0.06 psu) and lower seasonal amplitudes (average of 0.45 ± 0.06 psu), due to ocean flushing (Supplemental Table S2).

### Dissolved oxygen

For dissolved oxygen, all sites were dominated by the first harmonic, with the dip in the summer concentrations due to the inverse relationship of solubility with temperature as well as annual factors such as air temperature, circulation, vertical mixing, air-sea gas exchange, photosynthetic oxygen production, and use of oxygen in decomposition (Figure 2; Kim et al., 2018). Additionally, benthic primary producers, such as seagrass meadows (Berg et al., 2019) and microalgae (McGlathery et al., 2001), can have important impacts on the fluctuations of water column oxygen concentrations that vary seasonally. Mainland sites generally had earlier to mid-year dates of minimum dissolved oxygen (average of 216.4 ± 8.6 year day) and lower minimum dissolved oxygen values (average of 5.54 ± 0.55 mg/L) and moderate seasonal amplitudes (average of 2.59 ± 0.16 mg/L) due to longer residence times and warmer temperatures (Supplemental Table S3; Safak et al., 2018). Ocean inlet and mid-lagoon sites had higher minimum values (average of 6.09 ± 0.6 mg/L) and moderate seasonal amplitudes (average of 2.56 ± 0.25 mg/L) due to more consistently cool temperatures and higher flushing rates (Supplemental Table S3; Safak et al., 2018).

### Chlorophyll-*a*

For $\log_{10}$(Chl), mainland sites generally exhibited a mix of annual and semi-annual seasonal harmonics, while ocean inlet and mid-lagoon sites were generally dominated by the first harmonic (Figure 2). Seasonal cycles of chlorophyll can be impacted by nutrient inputs, temperature, light availability (del Carmen Jiménez-Quiroz et al., 2021), as well as tidal mixing, seasonal winds, upwelling, and stratification (Robles-Tamayo et al., 2020). In the VCR lagoons, the concentrations seem to be most impacted by hydrology, such as the addition of nutrients through groundwater and atmospheric deposition (McGlathery et al., 2007), which due to the shallow nature of coastal lagoons, tended to stay for longer, and could cause lingering elevated algal concentrations (Anderson et al., 2003, 2010; Tyler et al., 2003; McGlathery et al., 2007; Gilbert et al., 2014). Sites exhibited geographic differences in the date for maximum, as mainland sites had an average of 214.8 ± 5.9 year day, and ocean inlet and mid-lagoon sites had a larger spread in dates with an average of 234.2 ± 29.4 year day (Supplemental Table S4). The mainland sites had statistically significant higher maximum values (average of 9.64 ± 2.37 µg/L, p = 0.034) and seasonal amplitudes

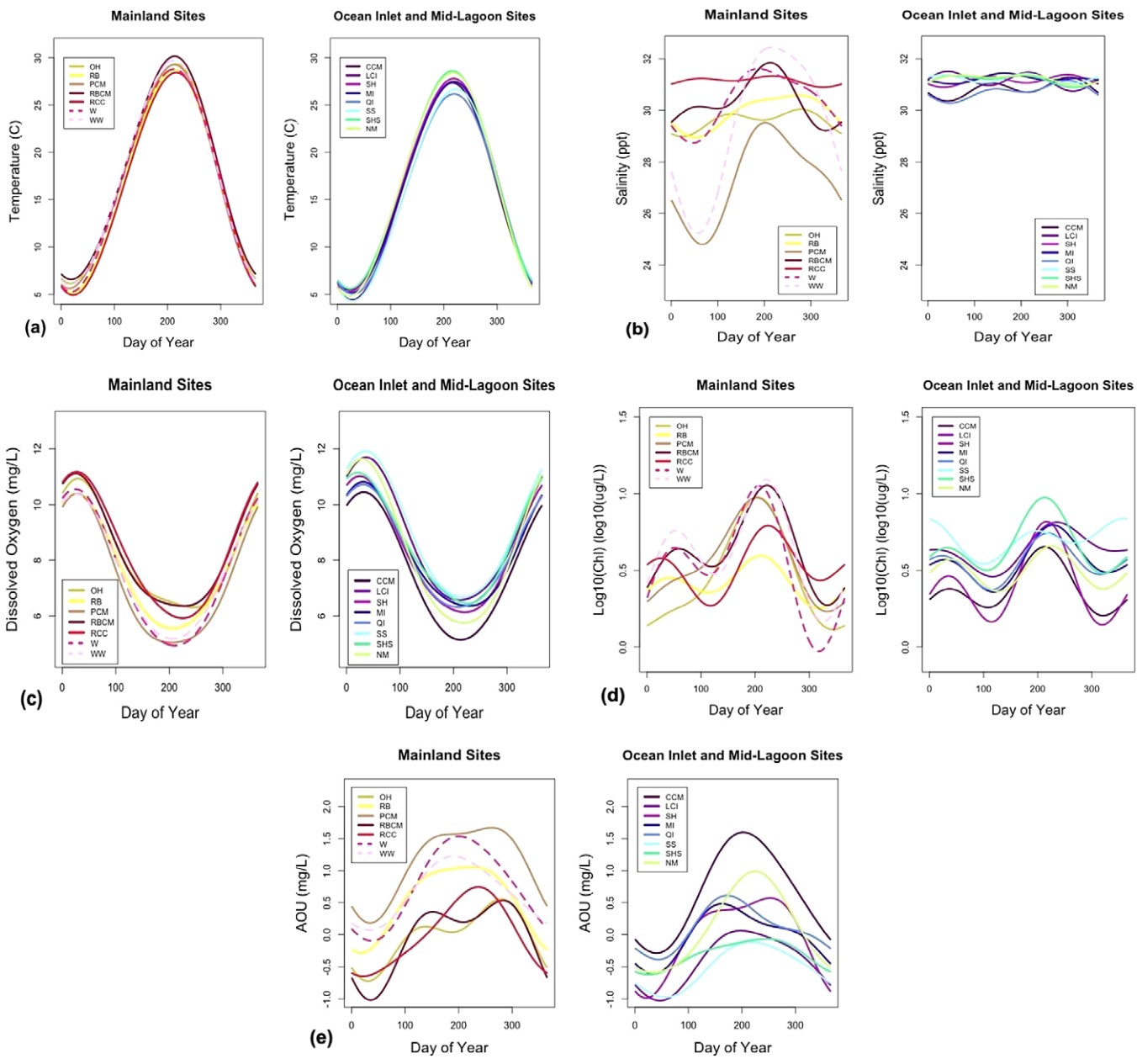

**Figure 2.** Temperature Composite harmonic fits for mainland sites (left panel) and ocean inlet and mid-lagoon sites (right panel for). (a) Temperature (b) Salinity (c) Dissolved Oxygen (d) Log10(Chl) and (e) AOU.

(average of 2.60 ± 0.51 µg/L, p = 0.009) likely related to longer residence times, allowing for more stagnant nutrient-rich waters (Supplemental Table S4; Safak et al., 2018). Ocean inlet and mid-lagoon sites had lower maximum values and seasonal amplitudes (averages of 6.24 ± 1.17 µg/L and 1.65 ± 0.17 µg/L), likely due to more mixing that dampens extremes (Supplemental Table S4; Safak et al., 2018).

### AOU

For AOU, all sites were dominated by the first annual harmonic (Figure 2). AOU is related to many of the same factors as dissolved oxygen, as well as biological activity (BCO-DMO, 2023). During summer, lower solubilities from warmer water temperatures drive lower dissolved oxygen (Boyer et al., 1999). In previous studies,

higher AOU values were found in the summer and fall months, with values closer to zero in the spring and winter when photosynthesis and respiration are low and roughly in balance and when air-sea gas exchange acts to resets AOU towards zero (Calleja et al., 2019). In general, mainland sites had later and more variable dates of maximum values (average of 240.8 ± 26.5 year day), while ocean inlet and mid-lagoon sites were earlier (average of 211.5 ± 19.2 year day) (Supplemental Table S5). The mainland sites had mid to higher maximum values (average of 1.13 ± 0.32 mg/L) and seasonal amplitudes (average of 0.77 ± 0.08 mg/L), indicating higher summer net community respiration, potentially related to proximity to marsh and organic inputs to the lagoon that fuel bacterial respiration (Supplemental Table S5; Ducklow and Doney, 2013). Values at ocean inlet and mid-lagoon sites were generally slightly lower (averages of 0.63 ± 0.40 mg/L and 0.70 ± 0.13 mg/L) where sites are

better flushed and away from marsh organic carbon inputs (Supplemental Table S5).

### Long term changes

Statistically significant long-term trends were found for only a subset of the water quality variables at some VCR LTER sampling sites (Figure 3 and Supplemental Table S7). There were no distinct trend patterns either by water quality variable or by geography, with statistically significant trends occurring for salinity, AOU, and $\log_{10}$(Chl) at both mainland and ocean inlet–mid-lagoon sites; the exception was temperature, which exhibited significant warming trends only at a cluster of mainland sites. Earth system modeling studies indicate the detection of ocean climate change trends requires 20–30 years for temperature and even longer (+50 years) for biogeochemical variables such as surface chlorophyll (Schlunegger et al., 2020). The absence of a statistically significant trend for many variables/sites (Supplemental Table S7) could reflect either a true lack of trend or detection issues because the signal-to-noise was small at sites sampled at low frequency with large seasonal and interannual variability (Henson et al., 2010); resolution of this issue will benefit in the future from longer coastal water quality time-series sampled at higher frequency.

Eight of the VCR LTER sites exhibited statistically significant, positive trends in $\log_{10}$(Chl) (Figure 3). Locations with significant positive $\log_{10}$(Chl) trends were split evenly between mainland ocean inlet and mid-lagoon sites in the northern portion of the VCR LTER sampling site, with no clear mainland-lagoon geographic pattern in trend magnitude (Supplemental Table S7). At the mainland northern VCR cluster of sites (RB, PCM, and RBCM), there was a co-occurrence of statistically significant, increasing temperature (mean $0.071 \pm 0.037$ °C/year) and $\log_{10}$(Chl) (mean $0.014 \pm 0.007 \log_{10}$(µg/L)/year) (errors on multi-site mean trends computed by propagating regression slope errors for each site

assuming regression errors are independent), consistent with warmer temperatures stimulating algal growth (Denchak, 2019).

Other findings from more nutrient-enriched Maryland/Virginia coastal lagoons included Chl increasing in the lower part of their study area, close to the northern edge of the VCR region (Wazniak et al., 2007; Gilbert et al., 2014). Southern Mid-Atlantic coastal bodies of water had consistently high eutrophic conditions, as well as elevated levels of chlorophyll-*a*, with coastal lagoons being more impacted than river estuaries (Bricker, 2007). Chl concentrations could also increase in the VCR area as macroalgal declines (McGlathery et al., 2001) or crashes, which can double the water column Chl concentrations (Tyler et al., 2001).

The changes in Chl were not limited to mainland sites, as $\log_{10}$(Chl) also exhibited statistically significant, positive trends at four ocean inlet and mid-lagoon sites (mean $0.017 \pm 0.005 \log_{10}$(µg/L)/year) (Supplemental Table S7). This could be related potentially to variations in water residence times at these sites due to changing hydrological factors (Denchak, 2019), though there is insufficient temporal information from simulated residence times, available for only two time periods, 2002 and 2009, to indicate any long-term trends (Safak et al., 2015; Safak et al., 2018).

Five VCR LTER sites exhibited statistically significant, positive trends in salinity (Figure 3). Salinity increased at three mainland sites (mean $0.085 \pm 0.040$ psu/year) and at two ocean inlet and mid-lagoon sites (mean $0.068 \pm 0.026$ psu/year). These trends could be related to decreased freshwater input due to droughts or low streamflow, higher inundation of sea level, or more intense storms causing breaching and washover events that can vary by site (Supplemental Table S7; Anthony et al., 2009). However, the underlying driving factors are difficult to reconstruct because there are no USGS gauged streams for the VCR LTER region and only a single NOAA tide station (Wachapreague, VA, ID: 8631044), north of the VCR LTER sampling region, where local sea-level increased

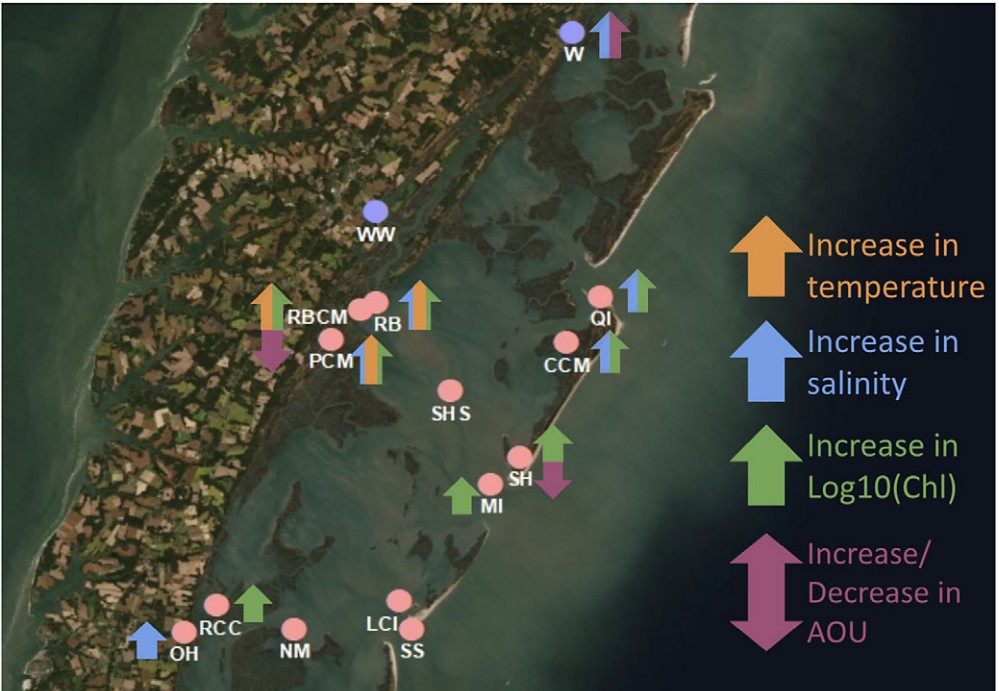

**Figure 3.** Spatial map of statistically significant multi-year temporal trends in water quality variables at VCR-LTER and ESL sites.

at 5.63 ± 0.59 mm/year over the past three and a half decades (NOAA, 2024). Coastal storms have been found to have large impacts on salinity, especially when storm surges and waves are more intense for storms occurring during high tide (Kurylyk and Smith, 2023).

Statistically significant temperature increases were limited to three previously mentioned mid-VCR mainland sites (mean 0.071 ± 0.037 °C/year), however, many other sites had positive, though statistically insignificant trends (Supplemental Table S7). Previous studies have found regional surface ocean and air warming trends and increased marine heatwaves likely associated with anthropogenic climate change (Wiberg, 2023). Including all VCR LTER sites including sites with statistically insignificant trends (Table S7) resulted in a regional-mean warming trend of 0.041 ± 0.019 °C/year that is broadly consistent with the measured trends (1982–2021) for nearby coastal ocean (0.030 ± 0.016 °C/year) and coastal bay (0.021 ± 0.015 °C/year) weighted to the Wachapreague site (Wiberg, 2023). Coastal water warming trends have been found to have a positive correlation with regional atmospheric and oceanic temperatures on both monthly and decadal time scales (Najjar et al., 2010).

Statistically significant negative AOU trends were found at two sites, one mainland and one ocean inlet and mid-lagoon site (−0.057 ± 0.054 and −0.044 ± 0.043 mg/L/year, respectively) (Supplemental Table S7). These AOU trends could be related to a change in lateral processes linking biogeochemical dynamics, organic carbon transport and freshwater flow (Figure 3). Murray et al. (2020) found that AOU is more influenced by the mixing of end-member waters with different AOU than within-estuary biology, which could help to explain our findings.

Temporal shifts in seasonal harmonic elements were found when the long-term VCR-LTER time series was broken into earlier and later time periods (Supplemental Tables S9 and S10). For temperature, dates of maximums generally shifted earlier with increased maximum values and seasonal amplitudes; the shifts in seasonal amplitudes were significant statistically for mainland and all sites. This aligns with shifting phenology due to global climate change (Anthony et al., 2009). For salinity, seasonal amplitudes increased with statistical significance at the mainland, ocean inlet-lagoon, and all sites. This aligns with the shift of salinity due to changes in freshwater hydrology, evaporation and runoff, and groundwater, and threat of saltwater intrusion from the nearby ocean (Anthony et al., 2009). For dissolved oxygen, dates of minimum values tended to shift earlier, while seasonal amplitudes increased, though neither of these signals was significant statistically for site types. The shift earlier of the dates of minimum value aligns with the shifting earlier of maximum temperature values. For $\log_{10}$(Chl), statistically significant increases in maximum values were found for ocean inlet-lagoon and all sites. Seasonal amplitudes also decreased with statistical significance at several sites, which could be related to continuously elevated chlorophyll concentrations due to potentially more stagnant waters and longer residence times (Bricker, 2007). For AOU, dates of maximum values shifted earlier (statistically significant for ocean inlet-lagoon sites), while seasonal amplitudes increased. AOU shifting earlier is likely related to the shifting of increased biological effects (Ganguly et al., 2015).

### Simulated low-frequency sampling

Sub-sampling experiments were conducted to evaluate the ability of the low-frequency VCR-LTER sampling to capture the climatological annual cycle. When randomly subsampling the sites with high-frequency data sampling at the rate of low-frequency mainland measurements, the resulting harmonic fits tended to show relatively low average nRMSE values, means of 0.692 ± 0.306 and 0.620 ± 0.220 for Wacharpreague and Willis Wharf, respectively (Supplemental Table S6). The relatively low nRMSE values indicated reasonable agreement between the low-frequency and high-frequency harmonic models.

Additionally, the low-frequency harmonics exhibited relatively good NSE values compared to the high-frequency harmonics, with averages of 0.751 ± 0.249 and 0.824 ± 0.172 for Wachapreague and Willis Wharf, respectively. The NSE values for all variables except for $\log_{10}$(Chl) were within the acceptable range (above 0.65) at Wachapreague, with temperature (0.991 ± 0.0005) and dissolved oxygen (0.943 ± 0.003) being very good (Supplemental Table S6). At Willis Wharf, the NSE values for all variables besides AOU were within the acceptable range, with temperature (0.992 ± 0.0004), salinity (0.943 ± 0.003), and dissolved oxygen (0.940 ± 0.003) all being well above the 0.65 threshold (Supplemental Table S6). $\log_{10}$(Chl) had the lowest average NSE at Wachapreague (0.296 ± 0.027) and highest nRMSE (1.217 ± 0.034), while AOU had the lowest average NSE (0.532 ± 0.027) and highest nRMSE (0.965 ± 0.025) at Willis Wharf, with NSE values for both falling below the adequate threshold (0.65) (Supplemental Table S6). Both AOU and $\log_{10}$(Chl) had slightly more complex seasonal cycles, with $\log_{10}$(Chl) having a mix of annual and semi-annual harmonics, as discussed above in the section "Chlorophyll-a" and Figure 2. Therefore, AOU and $\log_{10}$(Chl) may have been harder to capture more precisely at subsampled rates. Overall, for most of the water quality variables, the nRMSE values were relatively low, and the NSE values, for the most part, were in acceptable ranges, indicating the skill of long-term VCR-LTER time-series in estimating seasonal cycles.

The second set of subsampling experiments with varying durations of quarterly sampling showed higher nRMSE values for 5–10 years of sampling, indicating that sites with minimal data availability would not resolve the climatological cycle well (Figure 4; Supplemental Tables S11 and S12). After 50 years of quarterly sampling, Wachapreague had nRMSE across all variables of 0.298 ± 0.181 and Willis Wharf had 0.198 ± 0.077. Across variables, sites, and harmonic elements, the duration of quarterly sampling to reduce nSD and reach the high-frequency confidence intervals varied, with the number of years to reach within the full harmonic confidence intervals ranging from 10 to 23 years. It is important to note that for sub-sampling of Wachapreague $\log_{10}$(Chl), harmonic elements' confidence intervals were only crossed at one point for the date and value of maximum, and the seasonal amplitude. This could be because of the routine incorrect values at the site that may have been missed by the outlier removing processes. Overall, sub-sampling at Wachapreague indicated the need for slightly higher number of years of quarterly sampling for the sub-sampled low-frequency and high-frequency confidence intervals of harmonic elements (17.2 ± 4.2 and 15.7 ± 3.0 years, respectively) compared to Willis Wharf (14.9 ± 4.6 and 17.2 ± 3.3 years, respectively) (Supplemental Tables S11 and S12).

Shorter durations of quarterly sampling were required for certain variables to meet the high-frequency harmonic confidence intervals, with an average value of 13.3 ± 3.6 years for dissolved oxygen at Wachapreague and 10.0 ± 7.9 years for temperature at Willis Wharf (Supplemental Tables S11 and S12). The longest duration of quarterly sampling to reach the high-frequency confidence intervals was 23.3 ± 11.3 years for salinity at Wachapreague and AOU at Willis Wharf, an average of 23.0 ± 6.3 years

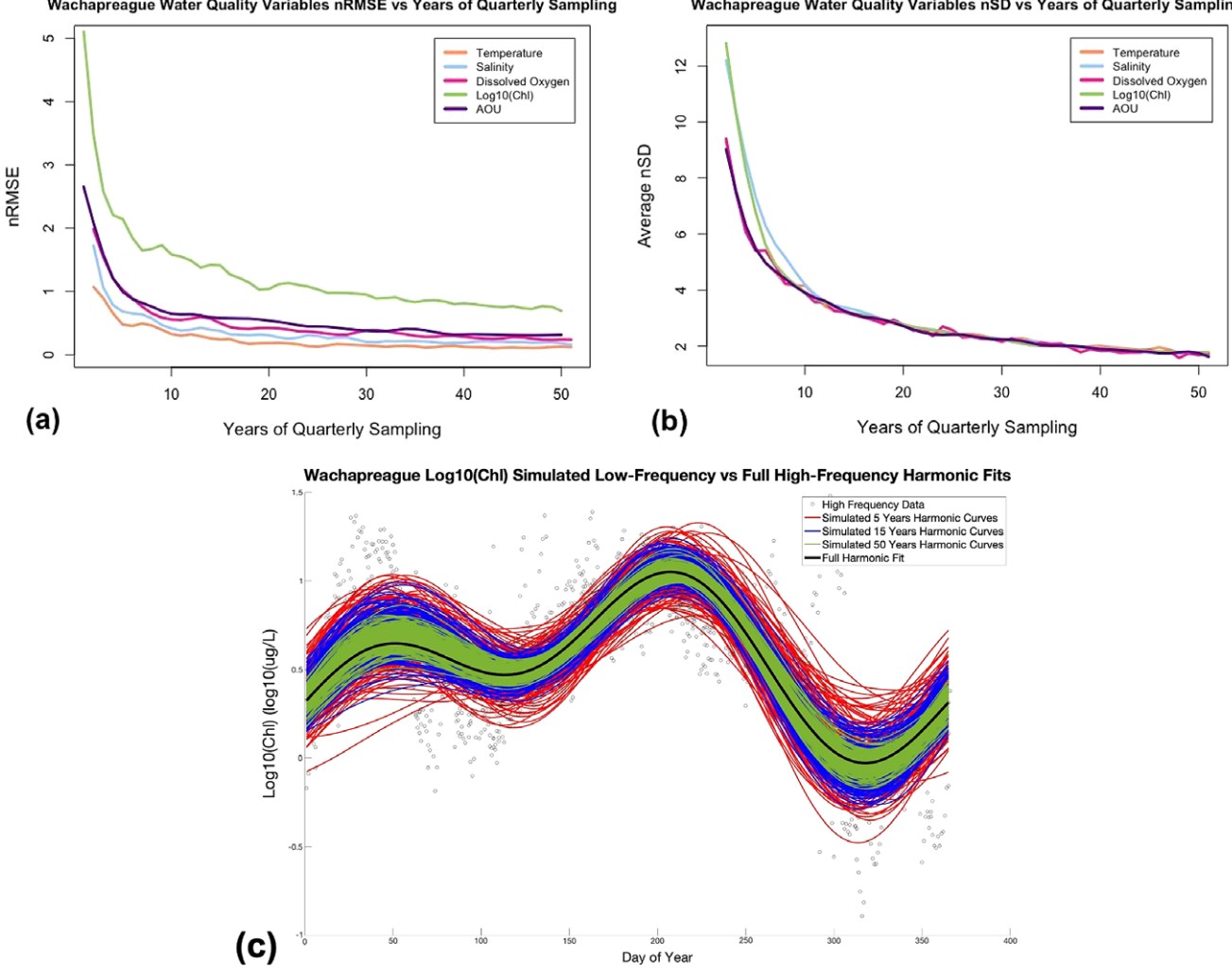

**Figure 4.** Influence of quarterly sampling duration on low-frequency sub-sampling harmonic fits versus high-frequency fits for (a) nRMSE, (b) average nSD of date of maximum value and seasonal amplitude versus successive years of quarterly sampling at site Wachapreague, and (c) harmonic curves for 200 trials of simulated low-frequency sampling of log10 (Chl) at site Wachapreague for sampling densities of 5 years (in red), 15 years (in blue), and 50 years (in green) versus the full high-frequency harmonic fit (in black) (data points marked as circles).

(Supplemental Tables S11 and S12). Wachapreague's salinity values could be related to the fact that it had a more complex seasonal cycle, and therefore, it is harder to capture in fewer years. Willis Wharf's AOU had the highest nRMSE compared to the full harmonic curve in subsampling, indicating that it takes more data points to accurately complete its harmonic curve (Supplemental Table S6).

## Conclusions

Using harmonic analysis, coastal water quality variables - temperature, salinity, dissolved oxygen, $\log_{10}$(Chl) and AOU - showed strong seasonality and geographic variation in the lagoon-barrier island system on the Eastern Shore of Virginia. The seasonal cycles of temperature, dissolved oxygen, and AOU in the VCR coastal lagoons were all dominated by an annual harmonic cycle, while salinity and $\log_{10}$(Chl) had a mix of annual and semi-annual harmonic cycles. Mainland sites generally had higher maximum temperature, $\log_{10}$(Chl), and AOU values, lower dissolved oxygen values, and more complex salinity seasonal cycles than ocean-inlet

and mid-lagoon sites due to longer residence times, shallower waters, and adjacence to marshes and uplands.

After the removal of these seasonal cycles, linear regression analysis showed that all the VCR coastal lagoon water quality variables exhibited significant long-term trends, at least at some sites, except for dissolved oxygen. Coastal water quality thus was not static to natural decadal variability and changing global climate conditions. Historical data analyses of multi-decadal time series, such as for the VCR coastal lagoons here, complement simulation-based approaches for determining the sample density and duration required to detect climate change signals. By dividing the VCR LTER time series into early and later time periods of roughly a decade each, changes in seasonal amplitude and phenology were identified from temporal shifts in some water quality harmonic elements. These shifts generally involved increases in seasonal amplitudes for most variables and earlier of dates within the year for seasonal minimums and maximums. As illustrated here, the coupling of harmonic and regression analyses provides a compact and consistent statistical approach for characterizing seasonal variability, long-term trends, and shifting phenology at coastal monitoring and research sites more generally.

The seasonal harmonics were captured relatively well for all variables when sites measured at a high frequency were subsampled at the equivalent seasonal resolution of the full VCR-LTER, low-frequency time series. This indicated that the multi-decade VCR-LTER low-frequency water quality sampling approach generated consistent and sufficiently dense data sets to estimate the seasonal cycles using harmonic analysis. Based on sub-sampling of high-frequency time series, the average duration of quarterly sampling needed to reach the confidence intervals for the water quality variables ranged from 10–23 years, indicating that below 10 years of low-frequency sampling, sampling is unlikely to resolve the climatological seasonal cycle. The nRMSE of these variables all plateaued by 50 years, indicating that 50 years of low-frequency sampling results in robust and accurate harmonic fits the climatological seasonal cycle.

The VCR-LTER project has long-term water quality data sets that range up to 30 years, within the estimated decade to multiple decade time window from the statistical sub-sampling of the high-frequency data. This suggests that the long data record from the VCR-LTER water-quality sampling scheme can be used to characterize well the seasonal variability across all variables investigated. A caveat is that long-term climate change trends may have already altered seasonal amplitudes and phenology, as suggested by statistically significant differences found for some harmonic parameters when the times series was sub-set into earlier and later periods.

The VCR coastal data analysis presented here illustrated the value of even a few years of high-frequency water-quality sonde data sets as a complement to more traditional and widely used manual data collection approaches. The information gained from coupling harmonic analysis with statistical sub-sampling of high-frequency records can guide researchers in analyzing existing and historical time series and establishing new water-quality monitoring sites. The statistical sub-sampling analysis highlighted clearly the trade-offs of sampling frequency versus duration or sampling density for identifying seasonal variation in the VCR coastal lagoons. The same approach can be applied more generally to other coastal sites and to other water quality variables that are not yet measured or just beginning to be measured at long-term sites sampled at a low-frequency, using a reference station measured at a high-frequency nearby if available.

**Open peer review.** To view the open peer review materials for this article, please visit http://doi.org/10.1017/cft.2024.6.

**Supplementary material.** The supplementary material for this article can be found at http://doi.org/10.1017/cft.2024.6.

**Data availability statement.** High-frequency water quality data were provided by the College of William and Mary's Virginia Institute of Marine Science Eastern Shore Laboratory (VIMS ESL) with the assistance of ESL's Darian Kelley. Data can be retrieved from the VIMS ESL website (https://www.vims.edu/esl/research/water_quality). Low-frequency water quality data (doi: knb-lter-vcr.247.17) were retrieved from the VCR website (https://www.vcrlter.virginia.edu/cgi-bin/showDataset.cgi?docid=knb-lter-vcr.247).

**Acknowledgements.** The authors would like to thank the VCR LTER and VIMS ESL teams who generated the extensive coastal water quality data sets used in this study. The authors also thank members of the Doney research lab for their support and guidance throughout the project.

**Author contribution.** E. I. B. initially devised the methodology, wrote the code and performed all of the analysis, and led the writing of the article. S. C. D. and K. J. M. contributed to method development, data analysis, and writing and editing of the article.

**Financial support.** E. I. B., K. J. M., and S. C. D. were supported by the University of Virginia, and we thank and acknowledge support from the VCR LTER project (National Science Foundation grant 1832221).

**Competing interest.** The authors declare no conflict of interest.

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
