## [Editor Report]

Dear authors

Please address the reviewers’ comments in detail; be specific about the objectives and research outcomes in the abstract. The second reviewer requests a renaming of stations, more general environmental conclusions and placing the results in better context.

---

## [Editor Report]

Thank-you authors for the revised manuscript.

Before acceptance there are some minor comments that need to be addressed – see editor comments and those of the Reviewer (inputs on revised article R1).

Handling Editor comments

Start first sentence of Abstract with a broad study objective that would be of interest to a large audience. For example …..

This study investigated data and trends in water quality to inform monitoring techniques and modelling at other coastal research sites.

Otherwise, details on Methods such as sensor calibration and cleaning for the sonde time-series have been completed. Also, the results and discussion were edited in this revision to infer more general environmental conclusions.

Because the Results and Discussion section is combined; the focus is very site specific. In the Conclusion a wider context is provided but please do see if you can address the reviewer’s concern “make the Introduction and Discussion more generalized rather than specific to your location”. This is a good study, but you need to highlight the significance of your results for others globally.

---

## [Editor Report]

The authors have attended to all editor and reviewer comments and the article is now ready for publication. Detail has been provided to highlight the significance of the study for others. Text has been adequately revised, long term changes described and some statistical explanations added (e.g. A new Table S10 was added to the Supplementary section to report results of a two-tailed unequal variance t-tests).